# Evaluating the Practicality of Counterfactual Explanations

**Nina Spreitzer**
University of Amsterdam
Amsterdam, The Netherlands
`nina.c.spreitzer@gmail.com`

**Hinda Haned**
University of Amsterdam
Amsterdam, The Netherlands
`h.haned@uva.nl`

**Ilse van der Linden**
University of Amsterdam
Amsterdam, The Netherlands
`i.w.c.vanderlinden@uva.nl`

## Abstract

Machine learning models are increasingly used for decisions that directly affect people's lives. These models are often opaque, meaning that the people affected cannot understand how or why the decision was made. However, according to the General Data Protection Regulation, decision subjects have the right to an explanation. Counterfactual explanations are a way to make machine learning models more transparent by showing how attributes need to be changed to get a different outcome. This type of explanation is considered easy to understand and human-friendly. To be used in real life, explanations must be practical, which means they must go beyond a purely theoretical framework. Research has focused on defining several objective functions to compute practical counterfactuals. However, it has not yet been tested whether people perceive the explanations as such in practice. To address this, we contribute by identifying properties that explanations must satisfy to be practical for human subjects. The properties are then used to evaluate the practicality of two counterfactual explanation methods (CARE and WachterCF) by conducting a user study. The results show that human subjects consider the explanations by CARE (a multi-objective approach) to be more practical than the WachterCF (baseline) explanations. We also show that the perception of explanations differs depending on the classification task by exploring multiple datasets.

## 1 Introduction

Machine learning (ML) models are increasingly used for automated decision-making impacting people's lives [28]. Some typical applications for ML model decisions are approving a requested loan [17], hiring an applicant [2], or setting the price rates for insurance contracts [27]. ML models are often opaque, meaning users cannot trace back how the decision is made [14]. In light of this automated decision-making, the European Union put forward a General Data Protection Regulation (GDPR) [1]. The GDPR includes a "right to explanation" [13], meaning affected people are entitled to request an explanation for a decision that has been made about them. To serve this right the research field of explainability for ML models is continuously growing [20]. Explainability aims to make the functioning of a model clear and easy to understand for a given audience [3]. However, an ongoing debate in legal and ML communities discusses what this right should entail and what specific requirements must be met [29]. Since the audience does not necessarily have technical skills

2022 Trustworthy and Socially Responsible Machine Learning (TSRML 2022) co-located with NeurIPS 2022.

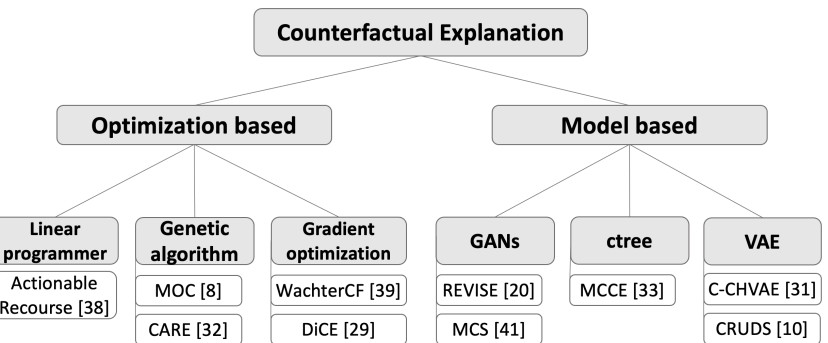

Figure 1: Some counterfactual explanation methods, categorized into optimization-based and model-based approaches.

or domain knowledge, explainability methods must also be suitable for non-technical users without expertise.

We focus on a particular method of post-doc explanations, called counterfactual explanations [30]. A counterfactual explanation proposes minimal changes to the input data that lead to a different model outcome. They can be seen as recommendations for what to change to achieve a desired model outcome [4]. We focus on two methods for computing counterfactual explanations, the original approach proposed by Wachter et al. [30], which we refer to as *WachterCF*, and a framework proposed by Rasouli et al. [24], called CARE. Counterfactual explanations are viewed to be easy to understand [30] and human-friendly [20]. Wachter et al. [30] state that counterfactual explanations are "practically useful for understanding the reasons for a decision". The Oxford Dictionary[1] defines practical as "concerned with the actual doing or use of something rather than with theory and ideas". Consequently, explanations must go beyond a purely theoretical concept to serve as practical explanations. However, there is limited work that attempts to evaluate the perception of counterfactual explanations in practice, and previous work has offered criticism on their applicability in real-life settings (see Section 2). Our contribution lies in first defining a set of properties that counterfactual explanations should satisfy in order to be considered practical. These properties are then used to define questions for a user study. The user study tests how users perceive counterfactual explanations computed by CARE and WachterCF in two different contexts. The research questions we explore in this work are as follows.

**RQ: How practical are counterfactual explanations for human subjects?**

Q1  What properties must counterfactual explanations serve to be practical for human subjects?
Q2  How do human subjects perceive counterfactual instances proposed by CARE compared to WachterCF?
Q3  How does the perception of counterfactual explanations differ depending on the classification task?

## 2   Related Work

In recent years, many methods for generating counterfactual explanations have been proposed [25]. A complete literature review of the proposed methods exceeds the scope of this paper; Figure 1 provides an overview. We distinguish between approaches that are optimization-based (based on an optimization problem) [7][21][24][28][30] and ones that are model-based (based on a machine learning models) [8][15][23][25][33]. In their work, Barocas et al. [5] show that the computation of counterfactual explanations often relies on widely overlooked assumptions that are necessary for counterfactual explanations to be accepted in real life. Laugel et al. [18] claim that assumptions make counterfactual explanations unreliable in many contextual uses. Researchers have responded to this criticism by developing counterfactual frameworks with different objective functions to satisfy specific properties [28][24][21][7][11][16]. WachterCF only focuses on being close to the original

---

[1]https://languages.oup.com/google-dictionary-en/

input data, but does not explicitly aim to satisfy any other practicality objective. In contrast, CARE solves a multi-objective problem including four desirable properties, such as *proximity* (being a neighbor of the ground-truth data [18]), *connectedness* (relationship between counterfactual instance and training data), *coherency* (keeping the consistency of (un)changed features) and *actionability* (include preference, e.g. restrictions or immutability of features). The following user studies have looked at counterfactual explanations. Warren et al. [31] have examined how well users are able to predict model outcomes after looking at counterfactual explanations. Förster et al. [11] examined the coherency of counterfactual explanations.

## 3 Methodology

We determine what properties counterfactual explanations must satisfy to be practical based on existing literature. We focus on properties that affect how human subjects perceive counterfactual explanations. The remaining sub-questions are answered by performing a user study with human subjects. The study aims to compare the perception of explanations provided by CARE with explanations provided by WachterCF. We use a non-parametric, called *Wilcoxon Sign Ranked* test [32] to determine if the results are significantly different. To measure the practicality of counterfactual instances, we formulate questions mapped to practicality properties. Before conducting a user study, the target group must be defined. Since any individual could be affected by automated decision-making, we use convenience sampling [10] by collecting information from closely available people. One distinction we make in the analysis is whether participants are familiar with machine learning methods because we believe there may be a potential difference in the perception of explanations depending on the prior knowledge of participants. We aim for 100 responses, preferably split evenly between respondents with a technical and non-technical background.

To examine different classification tasks, we use two separate datasets, bot accessible through the UCI machine learning repository [9]. The Adult Income dataset [12] is used to classify whether an individual is likely to have an income of more than 50k per year, whereas the Student Performance dataset [6] is used to predict if a student will pass a course. The two scenarios were evenly distributed among the participants. The preparation of the Adult dataset follows Zhu [34]. For the student dataset, we follow a similar structure. An overview of the final datasets can be find in the Appendix. For the remaining steps until the computation of the counterfactual explanation, we follow Rasouli et al. [24]. The data sets are split into 80% training and 20% test set. We use the same classification model as Rasouli et al. [24], which is a multi-layer neural network. Based on that, we compute the counterfactual instances using WachterCF and CARE. CARE provides the user with the possibility of defining constraints for actionability. We pre-define these constraints following Rasouli et al. [24] by setting gender and race as *fix* (immutable value) and age as *ge* (can only be greater or equal to the current value).

## 4 Practicality

Considering that the recipient of the explanation is a human subject, it is crucial to make the explanations human-friendly. Miller [19] summarizes human-friendly characteristics. Concerning counterfactual explanations, we have defined the following set of properties that lead to practical explanations. This set is used as a benchmark to evaluate how humans perceive counterfactual explanations in practice.

- **Contrastiveness**: Humans are not interested in why an event happened but rather in why that event happened instead of another. In the context of counterfactual explanations, we measure contrastiveness by how well the user understands what needs to change to get the opposite outcome.

- **Selectivity**: Generally, humans do not expect a complete cause of an event. Humans are used to selecting a smaller set of causes and treating it as a complete explanation. Therefore, counterfactual explanations can provide selectivity by changing only a subset of features as well as providing different suggestions.

- **Social**: The explaining method is part of an interaction between the end-user and the system explaining. As a result, the social environment, the target audience, and the use case need

to be considered. For counterfactual explanations, this means that the proposed changes should be made realistically in the given use case of the affected person.

- **Truthful**: A human-friendly explanation needs to make sense. In other words, the user must perceive the counterfactual suggestions to reach the other result as plausible.

- **Consistent with prior beliefs**: As described by the confirmation bias [22], people tend to ignore information that is inconsistent with their prior beliefs. Applied to counterfactual explanations, this means that end-users are more likely to consider explanations that suggest changes that are expected in advance.

## 5 Experimental Setup

We specify questions to measure how well the counterfactual instances satisfy the practicality properties. We map the questions shown to the properties as we believe they represent expectations of counterfactual explanations with respect to those properties. The person represented by the input data is called *Charlie*.

Table 1: User Study Questions: The following questions are formulated based on the property set to evaluate practicality.

| Question | Measurement | Property |
|---|---|---|
| **1** What attribute(s) would you expect to change for Charlie to instead get the outcome of "earning above 50k" / "passing the course"? | Multiple Choice: List of features | Consistency prior beliefs |
| **2** How surprised are you with the suggested changes in attributes to get the outcome of "earning above 50k" / "passing the course"? | Likert Scale: 1. Not at all 7. Very surprised | Consistency prior beliefs |
| **3** How well does the method explain to you what Charlie needs to change to get "earning above 50k" / "passing the course"? | Likert Scale: 1. Not at all 7. Very well | Contrastiveness |
| **4** Based on the explanation, what attribute(s) would you consider as most important to change the model outcome? | Multiple Choice: List of features | Consistency prior beliefs |
| **5** In your opinion, the amount of five different suggestions / one suggestion is ___ to explain the model outcome. | Single Choice: Too little/ Enough/ Too many | Selectivity |
| **6** In your opinion, the variation of attributes in the suggestion(s) is ___ to explain the model outcome. | Single Choice: Too little/ Enough/ Too much | Selectivity |
| **7** Do you think Charlie could realistically act upon the suggestions to change the model outcome to "earning above 50k" / "passing the course"? | Likert Scale: 1. Not at all 7. Fully | Social |
| **8** Do you think the suggestions make sense in order to retrieve the model outcome to "earning above 50k" / "passing the course"? | Likert Scale: 1. Not at all 7. A lot | Truthful |
| **9** Which method would you prefer as an explanation for the outcome of the ML model? | Single Choice: Method A Method B | |

The study starts by introducing ML models, counterfactual explanations, and the underlying classification task. For this study, 30 different instances from the two datasets are evenly randomized among the participants. The participants are asked what properties they expect. This is followed by showing the first counterfactual explanation computed by CARE and questions 2-8. After answering the questions, they are shown the explanation computed by WachterCF and are asked the same questions. Finally, the last question asks which method is preferred and gives an opportunity to leave a comment. The final user study shows each participant the CARE explanations first. Another option would be to randomize the order of explanation methods shown between participants. We chose the fixed order with the reasoning that our focus is on evaluating CARE (a method intended for practicality) against the baseline WachterCF method. By showing CARE first we ensure that CARE is evaluated without

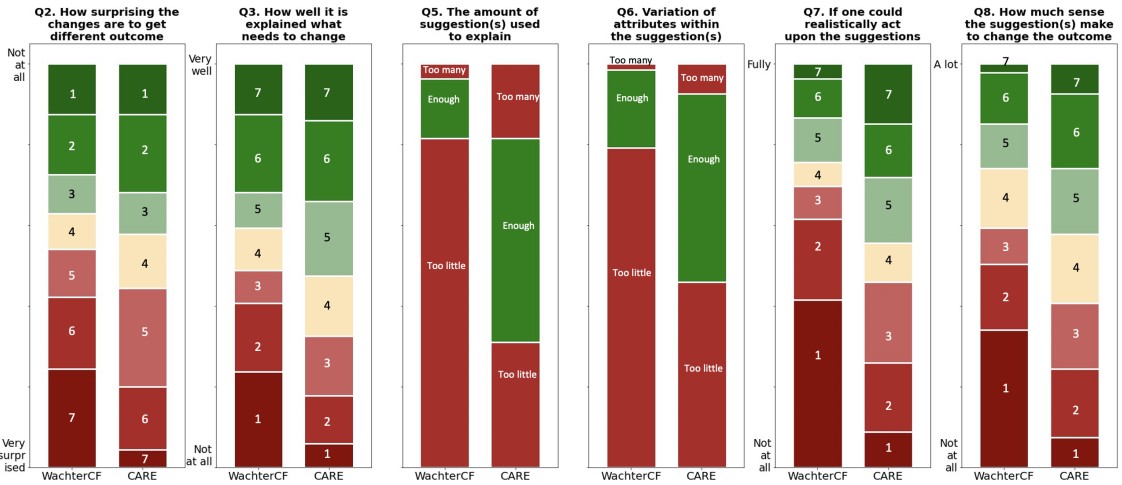

Figure 2: Responses to questions with quantitative responses (Likert scale and Single Choice). Red-colored answers (bottom) indicate negative responses, yellow (middle) neutral, and green positive (top). For each question, the responses for WachterCF are compared with CARE. All responses differ significantly between the methods.

any influence. If CARE is not perceived to be more practical than the WachterCF explanation shown second should obtain similar responses.

# 6 Results

Out of 135 complete responses, 69 participants received the Adult Income dataset, and 66 received the Student Performance dataset. Additionally, 70 responses indicate that they are familiar with ML models, while the remaining 65 imply not being that familiar with ML models. During our analysis, we do not find any significant difference in the perception of counterfactual explanations based on users' technical literacy. Figure 2 shows a stacked bar chart for Questions 2,3,5,6, and 7, comparing the responses of WachterCF and CARE. Looking at the graphs, it is noticeable that the counterfactual explanations calculated with WachterCF received more negative responses than those calculated with CARE. The *Wilcoxon Sign Ranked* confirms that the responses for WachterCF are significantly different from the responses for CARE. From this, we can conclude that counterfactual instances computed by CARE are considered more practical. Taking those results into account, we map back to the defined set of practicality properties. Question 2 shows that CARE provides less surprising explanations than WachterCF and is thus more *consistent with prior beliefs*. Question 3 shows CARE provides a better explanation of what needs to be changed to get the different model outcome, which maps to the attribute *Contrastiveness*. Questions 5 & 6 show that CARE is rated to *select* a better subset of features than WachterCF. Question 7 evaluates a *social* perspective, showing that CARE serves more realistic suggestions considering the specific use case. Question 8 shows that suggestions by CARE are perceived to make more sense than suggestions by WachterCF, which maps to being *truthful*.

By analyzing Questions 1 and 4 we can compute another estimator for being *consistent with prior beliefs*. Question 1 asks what features participants expect to change. Question 4 elaborates on what features are considered important after seeing the explanations. We analyze the responses by computing a *percentage of agreement*, which reveals how many of the features considered as most important after seeing the explanation were also selected in Question 1. The results show a slight tendency that explanations provided by CARE have a higher *percentage of agreements* than ones by WachterCF. However, according to the *Wilcoxon Sign Ranked* test, we do not have enough evidence to show a significant difference for this comparison. Question 9 directly asks the participants what method is preferred as an explanation. Out of the 135 answers, 113 selected CARE over WachterCF. Therefore, we can conclude that humans subjectively prefer CARE over WachterCF.

The responses show that the perception of WachterCF and CARE differs depending on the classification task. To further assess whether this difference influences the obtained results, we perform the same analysis as before but split the responses according to the classification task assessed by the users. The Student Performance data shows a similar pattern as the overall results shown in Figure 2. CARE is seen as more practical compared to WachterCF and all differences are statistically significant. On the contrary, the responses to the Adult Income dataset do not show the same results (see Appendix). Only Questions 5 & 6 show a significant difference in answers, which ask to indicate the perception regarding the practicality property *Selectivity*. All other questions do not differ significantly. Question 7 even shows a slight tendency in favor of WachterCF. We propose three reasons for this. First, when looking at the actual explanations, WachterCF often only changes the attribute *Age* in both datasets. If so, *Age* decreases for the Student Performance dataset, but increases for the Adult Income dataset. Decreasing the age is not an actionable explanation, but getting older is happening without any active changes. Therefore, seeing a decrease in age is unsatisfying. Another reason is that the student performance dataset contains more attributes to be modified than the adult income dataset. More attributes may lead to more complexity in computing practical counterfactual explanations. Furthermore, the decision of the Student Performance dataset is about whether a student is likely to pass or fail a class is a high-impact decision compared to the Adult Income decision. Participants may be more demanding regarding the practicality of the explanations when students' lives are directly affected.

## 7 Discussion and Future Work

In this section, we discuss the limitations and weaknesses of the methodological design and reflect on the results of the user study transitioning to possible future work.

One limitation of the user study is that the user study participants were selected through convenience sampling [10]. This type of sampling is prone to bias and could lead to a judgment that may not represent other social groups. Furthermore, the user study design does not represent a realistic scenario. Participants judge cases of people they do not know and to whom they have no emotional attachment. Another aspect of this is that the datasets have been preprocessed before conducting the user study, which may not be the case in a real-life scenario.

In this paper, we show that the perception of counterfactual explanation is different for two scenarios. This may suggest that the practicality depends not only on the computational method, but also on the type of ML task (e.g., the decision to be made), the data used to make the decision, and the complexity of that decision. In order to draw conclusions from this, further research is needed to explore practical use cases for which counterfactual methods can be used. This can be done by investigating how the perception differs depending on different decision types or the number of features. In addition, it is interesting to address how the results differ if the users themselves were affected by the decision, rather than evaluating an explanation for a stranger. Another area to explore further is to use this framework to compare user perceptions among different explanation methods, such as feature attribution methods (like LIME [26]) or causal explanations [31].

## 8 Conclusions

People affected by a decision made by an ML model have the right to receive explanations to understand better why and how the model makes a particular decision. Counterfactual explanations are a way to provide transparency by showing which and how attributes must change to achieve the desired outcome. Research has focused on developing various frameworks for computing counterfactuals. However, counterfactual explanations must be practical to be used by human subjects in practice. To answer our research question about the practicality of counterfactual explanations, we first define the following properties to measure practicality: contrasting, selective, social, truthful, and consistent with prior beliefs of the user. To test how people perceive the explanations and simultaneously answer the second and third sub-question, we conduct a user study to compare this method against WachterCF as a baseline using two different classification tasks. The overall responses show that people perceive explanations computed with CARE as significantly more practical than those computed with WachterCF. Furthermore, we analyzed that looking at both used datasets separately results in different outcomes, which indicates that the perception of an explanation might differ depending on the classification task, the data, or the scenario.

## Acknowledgments

We are grateful to the participants of the user study. And we thank Emma Beauxis-Aussalet (Vrije Universiteit Amsterdam) for the thoughtful feedback which improved the findings of this paper.

## Reproducability

We provide a repository containing the code and data that was used for this research. It also includes further details on the user study design. The repository is available at `https://github.com/ninaspreitzer/practicality-counterfactual-explanations`.

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

# 9 Appendix

**Results based on the Adult Income dataset**

Responses to Questions with quantitative responses (Likert scale and Single Choice) exclusively for the classification task of the Adult Income dataset. Red-colored answers (bottom) indicate negative responses, yellow (middle) neutral and green positive (top). For each question the responses for WachterCF is compared with CARE. Based on the P values of a Wilcoxon Sign Ranked only Questions 5 & 6 are significantly different. The P value for each question shown is accordingly: 0.95, 0.2, 0.001, 0.01, 0.5, 0.8

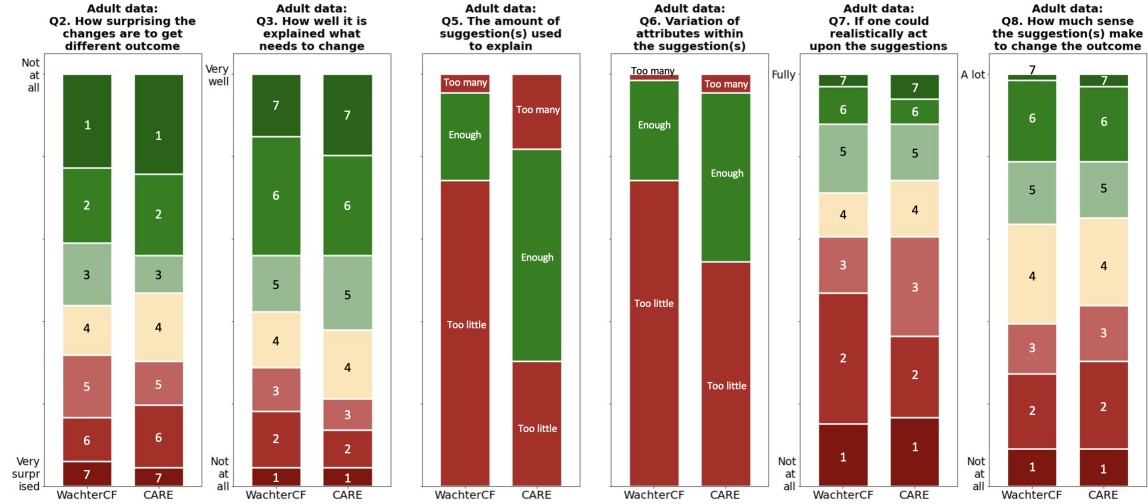

**Overview of preprocessed Adult Income dataset**

| Column | Type | Values |
|---|---|---|
| Age | Continuous | 17 - 90 |
| Working Hours | Continuous | 2 - 99 |
| Gender | Discrete | Female
Male |
| Race | Discrete | White
Other[2] |
| Education Level | Discrete | Less than High School
High School Graduate
Some College
Associate's Degree
Bachelor's Degree
Master's Degree
Doctoral Degree
Professional Degree |
| Marital Status | Discrete | Single
Married
Separated
Divorced
Widowed |
| Occupation | Discrete | Blue-Collar
White-Collar
Professional
Sales
Service
Other/Unknown |
| Industry Type | Discrete | Government
Private
Self-Employed
Other/Unknown |

**Overview of preprocessed Student Performance dataset**

| Column | Type | Values |
|---|---|---|
| Age | Continuous | 15 - 19 |
| Absences | Continuous | 0 - 30 |
| Gender | Discrete | Female
Male |
| Extra educational support | Discrete | Yes
No |
| Family educational support | Discrete | Yes
No |
| Paid tutor classes | Discrete | Yes
No |
| Study Time | Discrete | Very low
Low
Medium
High
Very high |
| Freetime | Discrete | Very low
Low
Medium
High
Very high |
| Going out | Discrete | Very low
Low
Medium
High
Very high |

