# OpenReview forum: "Evaluating the Practicality of Counterfactual Explanations"
_NeurIPS.cc/2022/Workshop/TSRML — TSRML2022_

### Official Review · Reviewer_T55i · 2022-10-18
**Review of Paper48 (lean towards accept)**

**Overall Rating:** 6

**Summary:**

The paper studies user perception of counterfactual explanations through a real-world user study that asks participants questions about two explanations (Wachter CF and CARE) under two different contexts (Adult Income and Student Performance). The questions are designed to reflect desirable properties that a CF explanation should satisfy as motivated by existing literature. The authors find that while participant perception of the two explanations differs in the Adult Income versus Student Performance setting, the participants generally preferred the CARE CF explanation over Wachter CF.

**Strengths:**

Overall, the paper has the potential to be a good piece of work in providing further insight into how everyday users might actually utilize CF explanations should they actually be implemented in real-world decision-making systems. I liked how the authors designed the questions that were asked in the user study around well-motivated properties from existing literature. The results also motivate the need for further study on user perceptions of CF explanations in more diverse settings beyond the two UCI datasets considered in this work.

The writing was also generally well-written and easy to follow.

**Weaknesses:**

The user study compares two explanations, but I wonder are these two explanations actually commonly used (note that the related work lists many CF explanation methods). What we have here is a result of how one explanation compares to another explanation; however, what we might care about is actually which explanation out of all possible CF explanations is the most "practical". I am left wondering after reading the user study what insights from the study may help us identify better explanations beyond the two considered.

Another main point about the organization of the paper: I found it difficult to understand how the properties mapped to the questions in Table 1 and would strongly advise reorganizing Sections 4 and 5 to better describe how different questions are motivated by the various properties of interest.

Other comments:
- While one can infer from Figure 1 what optimization-based and model-based mean, it would be better to define them explicitly in the text.
- I would also recommend taking a look at "Why Am I Not Seeing It? Understanding Users’ Needs for counterfactual Explanations in Everyday Recommendations" (Shang et al., 2022), which also ran a user study to understand the practicality of CF explanations.

**Overall Recommendation:**

I would lean in favor of recommending that this paper is accepted but would strongly urge authors to revise the paper to address the weaknesses mentioned above.

Note that there is a link in the paper to the author's personal GitHub which violated the double-blind guidelines.

**Review Confidence:**

4: The reviewer is confident but not absolutely certain that the evaluation is correct

---

### Official Review · Reviewer_Sdhv · 2022-10-19
**Review of Paper48 (Weak Accept)**

**Overall Recommendation:** Weak accept.
**Overall Rating:** 6

**Summary:**

This paper evaluates whether or not counterfactual explanations (CEs) are practical to end users based on various criteria drawn from the social sciences. It compares one of the first CE works, Wachter, with a newer method CARE.

**Strengths:**

The paper is mostly well written, and very readable. It provides a clear structure for evaluating the practicality of CEs and this can help other works to strengthen their own user studies, which tend to be rather sparse/inconclusive. While the framework deployed may lack some details, it is mostly rigorous and a good starting point. Most conclusions are justified well.

**Weaknesses:**

The paper perhaps lacks references to other works which evaluate practicality of CEs in other ways e.g. https://arxiv.org/abs/2203.06768 .

Although many ideas are listed, there seems a limited scope for immediately impactful future work or improvement. I enjoyed the paper nonetheless.

Good work has been done, but I would think that the novelty is limited given it is more a combination of prior works.

I would say that the method proposed in Wachter is known to have many shortcomings, being one of the first proposed CE works, making some of the conclusions unsurprising. It would be good to compare possibly 2+ more recent methods which aim to tackle specific CE issues, aligning each of their strengths with the evaluation criteria you give, perhaps.

The references notation seems a bit messy where names occur multiple times in one reference- I am not sure if this is convention but I would recommend changing to a simpler (e.g. numbered) system.

Typo on line 167 "consist".

**Review Confidence:**

3: The reviewer is fairly confident that the evaluation is correct

---

### Decision · Program_Chairs · 2022-10-23

Accept